# Visual Noise Mask for Human Point-Light Displays: A Coding-Free Approach

**DOI:** 10.3390/neurosci6010002

**Published:** 2025-01-02

**Authors:** Catarina Carvalho Senra, Adriana Conceição Soares Sampaio, Olivia Morgan Lapenta

**Affiliations:** Psychological Neuroscience Laboratory, Psychology Research Center, School of Psychology, University of Minho, Rua da Universidade, 4710-057 Braga, Portugal; catarinacarvalhosenra@gmail.com (C.C.S.); adriana.sampaio@psi.uminho.pt (A.C.S.S.)

**Keywords:** point-light walkers, visual noise, action perception, Blender software

## Abstract

Human point-light displays consist of luminous dots representing human articulations, thus depicting actions without pictorial information. These stimuli are widely used in action recognition experiments. Because humans excel in decoding human motion, point-light displays (PLDs) are often masked with additional moving dots (noise masks), thereby challenging stimulus recognition. These noise masks are typically found within proprietary programming software, entail file format restrictions, and demand extensive programming skills. To address these limitations, we present the first user-friendly step-by-step guide to develop visual noise to mask PLDs using free, open-source software that offers compatibility with various file formats, features a graphical interface, and facilitates the manipulation of both 2D and 3D videos. Further, to validate our approach, we tested two generated masks in a pilot experiment with 12 subjects and demonstrated that they effectively jeopardised human agent recognition and, therefore, action visibility. In sum, the main advantages of the presented methodology are its cost-effectiveness and ease of use, making it appealing to novices in programming. This advancement holds the potential to stimulate young researchers’ use of PLDs, fostering further exploration and understanding of human motion perception.

## 1. Introduction

Point-light displays (PLDs) are widely used stimuli to investigate human movement recognition and processing. Since the pioneering work of Johansson in 1973 [1], this technique has been highly applied in several domains, including biomechanics, motor, and cognitive neuroscience [2]. These stimuli are generated by filming human actors wearing dark suits with reflective patches at their major joints [1,3,4]. Action understanding is crucial for human interactions [5].

The neural processing pathway of point-light actions involves multiple stages, from visual input to motion control. Initially, the eyes process and transmit the visual information to the primary visual cortex (V1) [6]. From there, the information flows through the dorsal and ventral visual streams, where the first identification of space position and the visual elements’ categorisation occur [7]. Subsequently, the superior temporal gyrus and inferior parietal lobe are activated [8], as are the ventral portion of the premotor cortex and inferior frontal gyrus [6]. These areas are part of the mirror-neuron system and are crucial for action understanding [9]. This activation indicates that even simplified point-light animations can recruit action observation networks. In fact, human PLDs have been shown to evoke similar corticospinal excitability to real human movement, suggesting their effectiveness in motor resonance and action observation tasks [10].

Importantly, this process is not strictly linear. There are interactions between the dorsal and ventral streams [11], and information processing occurs in parallel across multiple brain regions, especially in relation to higher-level processing during action identification.

In sum, the neural processing of point-light actions involves a distributed network of brain regions, from early visual areas to higher-level regions responsible for action understanding, representation, and anticipation. This pathway demonstrates the brain’s ability to extract meaningful information from sparse visual inputs and link it to motor representations and action understanding.

Considering that humans are experts in recognising and predicting human motion, Cutting and colleagues [12] suggested adding visual noise to mask PLD stimuli, thus increasing task complexity. They tested static, dynamic, and linear masks. The static mask consisted of additional stationary dots; for the dynamic mask, the noise dots randomly changed on each frame; and the linear mask consisted of the dot elements moving in a specific orientation [12]. The static and linear masks were ineffective in disturbing the identification of the human agent and its action compared to the no-mask condition. In contrast, the dynamic mask decreased performance compared to all other conditions, making it harder for participants to identify the human agent and its movement. Unfortunately, this report provides little information on how to construct masks and only presents general concepts of movement type and applied velocity [12].

Subsequent studies have adopted masking strategies using additional moving dots (e.g., [13,14]. Decatoire and colleagues [9] developed PLAViMoP (Point-Light Display Visualisation and Modification Platform), a software containing two versions (1 and 2) for visualising and transforming 3D point-light sequences. The platform offers an interesting approach that allows users to add certain types of masks to their PLD sequences; however, it has some constraints, such as requiring specific software and file formats [15].

Research constantly needs diverse stimuli material [16], and thus, several techniques have been proposed to create PLD stimuli and masks [3,16,17,18]. Still, there is a lack of guidelines, as most available materials do not provide step-by-step explanations and sometimes do not specify the software used [12,19], which hampers the consistency and reproducibility of noise mask development. Furthermore, most procedures demand specific and paid software that usually supports specific file formats used in biomechanics [15]. Additionally, they require users to have programming skills if there is a need to customise the mask to the desired outcome [15].

To address these issues, we describe a step-by-step process for creating and adapting visual noise masks for PLDs using free, user-friendly software that requires no programming experience, as it can be operated solely via a graphical interface. This method aims to facilitate the creation of PLD stimuli and masks for researchers and students at any career stage, making the process much less time-consuming and offering a faster learning process than programming. Even in informatics technology, novice programmers face a long-term learning curve before having a solid grasp on this subject [20].

To achieve our goals, we selected a free software commonly used for game development [21] that allows *playing* with many properties that others do not. Specifically, it supports various file formats and allows customisation of frame rate, texture, colour, and opacity. It offers a graphical interface, making it accessible without prerequisites. In sum, the major advantages identified were (a) the simplicity and accessibility of the software itself, (b) the ease of modifying and manipulating the parameters of the masks, and (c) the possibility to create the stimuli from scratch. Therefore, we detail the construction of two PLD noise masks with 22 and 33 dots as well as how to merge them with PLDs and, finally, present the results from a pilot experiment testing its efficacy on challenging stimuli recognition.

## 2. Materials and Methods

### 2.1. Step-by-Step Procedure for Building the Noise Mask

For creating our stimuli, we used Blender software 2.91.2 (Blender Foundation and Institute, Amsterdam, The Netherlands [21]), which supports a wide range of file formats, such as OBJ, FBX, 3DS, PLY, and STL, to be imported and exported.

Once in the software environment (available for download at https://www.blender.org, accessed on 31 December 2024), the first step is to choose either the 2D or 3D option according to specific needs and objectives. Note that the software supports the entire 3D pipeline, including modelling, rigging, simulation, rendering, compositing, motion tracking, and video editing [21]. The 2D animation pipeline also includes features such as colouring, animating, applying modifiers, cut-out animation, motion graphics, and grease pencil tools [21].

Our project used Blender’s 2D feature to align with the biological PLD data from Lapenta and colleagues’ [4] database. Please note that most 2D instructions can also be applied when using 3D options. Finally, we used the Blender editing header menu to overlay the human PLD stimuli with the newly created mask.

Next, we selected the *Draw Mode* to establish the desired parameters for the masking dots. We then chose the *Drawing a Circle* option, drew with the mouse, and pressed the ‘Enter’ key once finished. Then, we switched to *Object Mode* to access a toolbar containing tools to check the drawn object’s measurements and set its displacement along the ‘x’ and ‘y’ axes.

In the right panel (the *Properties Editor*), we set the background colour to black by clicking on *World Properties*, which enables background settings. We also selected the stroke colour as white under *Material Properties*, where the base colour, shape, or mode for any drawing generated over the background can be defined. Additionally, we specified the stroke texture as solid, with opacity set to 1.

Finally, we assigned half of the dots a size of 0.100965 m and the other half a size of 0.060999 m. These sizes were determined through trial and error. Initially, the dots in the original stimuli [4] were measured to determine the average size of the visible points. Based on these measurements, the sizes of the mask dots were adjusted by testing slightly smaller and larger values until the desired visual configuration was achieved. We chose these two sizes to reflect slight variations in dot size within and between the dots of our PLD stimuli. These variations occur due to the agent’s movement in space, as the PLD was created using human participants wearing reflective patches [4]. Minor changes in dot size arise based on the distance from the camera: for instance, in a walking movement, dots gradually increase in size as the agent moves closer to the camera. Similar size changes occur in movements involving the legs or arms, such as a low kick or forearm motion. Once the colour, opacity, size, and texture were set, the dots were copied in the *Scene Animation* panel to organise the quantity of dots and their initial positions on the screen, specifically the displacement of 22 or 33 dots.

Once the dots were ready, we created their dynamic movement, a key property that, in a 2D environment, relies on adjusting an object’s position along the ‘x’ and ‘y’ axes. To change a dot’s position, we accessed the toolbar, selected the *Move* option, and adjusted the dot along the ‘x’ and ‘y’ axes. For smooth movement, it is essential to change the position gradually, either on every frame or across a sequence of frames, depending on the original PLD videos and the desired speed for the mask. For our masks, the ideal speed to match our stimuli was defined as a displacement every 7 frames.

To set these parameters, we switched between the *Object* and *Edit* modes or clicked the *Edit* button available for each object on the view layer (green button). A helpful practice to prevent dots from swapping positions instead of moving smoothly is to use the *Interpolate Sequence* command at the top of the workspace. This command should be applied at every frame interval and to each dot.

Similar to the size of the dots, the movement intervals were established by observing the motion dynamics in the original videos, ensuring that the mask dots blended naturally with the video velocity while avoiding overlap or replication of the original movement. The mask aimed to create enough disruption to make recognising the motion more challenging. This approach was primarily guided by qualitative analysis, with the goal of balancing interference in the stimulus.

It is advisable to save progress at each step using the *Save As* option, which outputs a Blender project file necessary for rendering the final video. To render the video, we accessed the *Rendering* menu, selected the *Render Animation* option, and configured settings such as the desired frames per second (see Figure 1 for a diagram of these steps).

The files for the masks described here and the Blender project are available for download as .zip archives in an online repository (https://osf.io/q2u95/, accessed on 31 December 2024).

### 2.2. Combining Videos with the Mask

Once the noise masks were prepared, we combined each of them with our PLD stimuli using the same software. Therefore, we used the *Video Editing* tool; by using this selection, the desired video and the mask only have to be dragged to the edition panel. Each video was selected individually, and we then added an *Effect Strip* with the *Transform* option (one of several available options). This allowed each mask and stimuli video to have a transform strip attached to it, enabling the integration of the mask video into the stimuli video.

Importantly, the original PLD did not undergo any enhancement or modification of its original properties. The PLD stimuli retained their inherent characteristics, such as dot sizes and brightness, without any alterations beyond the addition of the noise mask.

Next, we selected the *Compositing* option (under the *Active Tool* option), which contains the *Blender* and *Opacity* properties necessary to adjust the parameters for combining the videos and creating the overlapping effect. For the mask video, we selected the *Alpha Over* blending option and set the opacity to 1. For the overlapping, we selected the blender option of *Alpha Over* for the mask videos and set the opacity to 1. For the stimuli videos, we selected the blender option of *Add*, and the opacity was also set to 1.

These options were chosen for the mask and stimuli because *Alpha Over* defines how to layer the mask on top of the video, while the *Add* option specifies which video will receive the mask. The opacity setting controls the transparency of the videos; our selection ensured that both the mask and stimuli videos were fully opaque with no transparency.

Once this process was complete, the video was rendered as described previously (see Figure 2 for an illustration of the final videos).

### 2.3. Testing the Masks

Once the videos were ready, they were tested to evaluate whether, as expected, adding dynamic masking with 22 and 33 dots to the PLD would increase the difficulty of stimuli detection. Therefore, we conducted a pilot experiment with 12 participants (6 male) aged 22–39 years (M = 33.42; SD = 5.18). Similar sample sizes have been adopted in experiments using this type of stimuli in action perception studies (e.g., [6,22,23]). The experiment followed the Declaration of Helsinki and was approved by the local Ethics Committee for Investigation on Social and Human Science of the University of Minho, Braga, Portugal (CEICSH 069/2020). Participants were recruited using a convenience sampling method through a network-based approach. The sample consisted of individuals from various professional backgrounds, including a pilot, a veterinarian, a fashion designer, a chef, two psychologists, two economists, a biologist, a tourism professional, a philosopher, and a hospitality professional. This diversity ensured that the participants were not solely from a psychology background, thereby minimising the risk of bias related to familiarity with PLD stimuli. All participants gave their written informed consent and complied with the following criteria: age between 18 and 45 years, normal or corrected-to-normal visual acuity, no known past or current psychological or psychiatric disorders, and no history of central nervous system injury.

The selected subset of 7 biological (Bio) and 7 scrambled (Scr) movements from Lapenta and colleagues [4], combined with the created masks (for details on the stimuli and the methods used for their creation and validation, please refer to [4]), were used in the experiment. Three conditions were generated for each stimuli type (Bio vs. Scr), specifically NoMask (original stimuli without visual noise), Mask22 (original stimuli + 22 dots mask), and Mask33 (original stimuli + 33 dots mask).

The task was built and presented using PsychoPy software (v2022.2.4) [24] and comprised a training phase with 6 trials (one for each stimuli condition combination) to familiarise participants with the task and clarify any doubts. Then, three experimental blocks were presented.

The first block (B1) contained 28 stimuli (7 Bio and 7 Scr from the Mask22 and Mask33 conditions), the second block (B2) contained the 14 NoMask stimuli, and the third block (B3) included all 42 videos. We chose this approach because most studies present only masked stimuli, and we wanted to investigate whether knowing the unmasked stimuli (i.e., after the NoMask condition) would affect detection. Stimuli were presented randomly within each block. Each trial began with a grey screen displaying a black fixation cross (400–900 ms), followed by the video presentation (~1000 ms), and so forth. Participants were instructed to respond as fast and accurately as possible during the video presentation, pressing one of two keys on the keyboard to indicate whether or not they detected a human action in the stimulus. The response was defined as accurate if participants defined a Bio video as containing human action and a Scr video as not containing human action.

## 3. Results

Our initial result was the generation of two types of dynamic masks, one containing 22 dots and another with 33 dots, whereas the second outcome consisted of testing them. These masks were subsequently applied to selected point-light stimuli, creating six experimental stimulus categories—BioNoMask, BioMask22, BioMask33, ScrNoMask, ScrMask22, and ScrMask33—to test their efficacy. The results from these tests constituted the second outcome.

To test the efficacy of these masks in obstructing stimulus identification, we first examined whether the presentation of unmasked stimuli affected the identification of masked stimuli. To this purpose, we conducted repeated measures analyses of variance (rmANOVA) for accuracy and reaction time for (i) unmasked stimuli (NoMask) identification, with block (B2 vs. B3) and stimuli type (Bio vs. Scr) as factors, and (ii) masked stimuli identification, with block (B1 vs. B3), condition (Mask22 and Mask33), and stimuli type (Bio vs. Scr) as factors.

The rmANOVA results showed no significant main effects for block or any interactions with other factors. Consequently, we analysed all trials from all three experimental blocks together. For all tests, we adopted alpha = 5%.

The accuracy data showed a normal distribution according to the Shapiro–Wilk normality test. The rmANOVA for accuracy, considering condition (NoMask vs. Mask22 vs. Mask33) and stimuli (Bio vs. Scr), revealed a main effect of stimuli (F1,11 = 8.29, *p* = 0.015, *ηp*2 = 0.430, BF10 = 43.51), which showed higher accuracy for biological versus scrambled PLDs, and of condition (F2,22 = 33.32, *p* < 0.001, *ηp*2 = 0.752, BF10 = 873.34) and stimuli*condition interaction (F2,22 = 11.43, *p* < 0.001, *ηp*2 = 0.510, BF10 = 335,484.32). Bonferroni-corrected post hoc analysis revealed higher accuracy for NoMask compared to Mask22 (*p* < 0.001, BF10 = 545.897) and Mask33 (*p* < 0.001, BF10 = 2264.784) but no differences between Mask22 and Mask33 (*p* = 0.777, BF10 = 0.339). Bonferroni-corrected post hoc for the interaction showed that accurate responses for the biological PLDs in the NoMask condition were higher than for all other stimuli: BioMask22 (*p* = 0.004), BioMask33 (*p* < 0.001), ScrNoMask (*p* = 0.003), ScrMask22 (*p* < 0.001), and ScrMask33 (*p* < 0.001). Statistical results are illustrated in Figure 3, and the mean accuracy for each stimuli type is depicted in Table 1.

For the reaction time analysis, we included only trials in which participants’ responses were correct. According to the Shapiro–Wilk normality test, all conditions except for ScrNoMask (*p* = 0.033) showed a normal distribution. Consequently, *p*-values are reported with the Greenhouse–Geisser correction applied. The rmANOVA for reaction time, considering condition (NoMask vs. Mask22 vs. Mask33) and stimuli (Bio vs. Scr), showed no main effect for stimuli (F1,8 = 0.134, *p* = 0.724, *ηp*2 = 0.016, BF10 = 0.270), condition (F2,16 = 2.520, *p =* 0.116, *ηp*2 = 0.240, BF10 = 0.249), or stimuli*condition interaction (F2,16 = 1.003, *p* = 0.377, *ηp*2 = 0.114, BF10 = 0.071).

To further investigate the efficacy of the masks in identifying a human agent in each of the biological actions, we summed the total number of correct answers for each movement across the 12 participants and calculated the percentage accuracy using the formula Nca/Tstim × 100 (where Nca = number of correct answers, and Tstim = total number of the specific biological stimuli in each condition, which is 24 for each action). As shown in Table 2, the identification of stimuli decreased with both masks for nearly all movements. Interestingly, for the Jump Rope action, the mask containing 22 dots reduced the accuracy to 25%, while accuracy remained the same for the NoMask and Mask33 conditions.

## 4. Discussion and Conclusions

This article proposes and explains a user-friendly approach for creating visual noise masks for PLDs. We chose Blender software [21], which allows the construction and modification of videos. The software seems valuable for researchers studying visual perception of animated actions, as well as for professional animators. It enables the creation of stimuli from scratch or the editing of previously recorded motion data and their masks. Another exceptional feature of this program is the ability to work with both 2D and 3D formats, offering built-in video editing tools. Furthermore, the software’s flexibility in manipulating both 2D and 3D file formats makes it particularly useful for editing body kinematics recorded with motion capture systems like Vicon [25,26] and for adapting noise masks accordingly. Although the software may encounter limitations with high-resolution animations or complex physics simulations, as rendering times can increase significantly in more demanding projects [27], its efficiency in generating simple visual stimuli makes it an excellent tool for our purpose.

We successfully built our noise mask for PLDs selected from a validated database [4] and described the methods to guide researchers in producing similar stimuli. Furthermore, we tested the efficacy of the masks in a simple forced-choice task, where participants had to indicate whether they could see a human action in both the unmasked and masked videos. Reaction time results revealed no significant differences between conditions, likely because our stimuli were presented for only 1 s, and responses were only registered if provided within this timeframe, thereby increasing the task’s difficulty. Indeed, the accuracy results revealed a statistically significant main effect for condition, showing that participants made more mistakes when the PLDs were presented with dynamic 22- and 33-dot masks. Errors included failing to detect human actions in biological motion, incorrectly identifying actions in scrambled motion, or not responding in time. This provides empirical evidence of the effectiveness of masking. Notably, when evaluating the percentage accuracy for each biological stimulus, there was a general decrease when presented with masks, except for the Jumping Rope stimulus in the Mask33 condition. It is important to emphasise that the experiment was designed to evaluate the masks’ effects at the condition level (i.e., NoMask, Mask22, and Mask33), as is typical in action perception experiments [6,10,22,23]. Investigating their effects on specific actions would require significantly more repetitions for each action type. Nonetheless, both masks demonstrated a robust general effect in reducing action identification, with statistically significant results and large effect sizes.

The findings suggest that the 22-dot mask is effective for this dataset and other similar ones. Furthermore, as there was no significant difference between the 22-dot and 33-dot masks, we consider the 22-dot mask effective for this dataset and other similar ones. While more complex masks with varying dots and motion trajectories could be an interesting direction for future research, designs with too many dots may overly obscure stimuli, making them virtually unrecognisable. Additionally, as standardisation within the study is essential, we opted to create a single motion trajectory, reinforcing the proposed approach’s practicality.

We have made our masks and the Blender project available to facilitate the work of investigators who need to use these procedures. In our Blender project, it is also possible to set a different amount of frames per second to each of the dots to depict their trajectory and speed, even though we did not use that feature because our goal was to have the same smoothness and speed for all dots. Additionally, it is possible to edit our project by copying or deleting the current dots to define the desired number for future experiments.

This editing flexibility is important because the configuration used herein was specifically tailored to the videos and database serving as the basis for this experiment. Thus, the parameters used in our experiment should not be considered universal for all types of PLD stimuli. It is important to emphasise that our primary goal was to present an alternative to programming for creating PLD noise rather than to develop a mask that fits all stimuli. While our approach is generalisable, we recommend adjusting the parameters as needed for different stimuli and research objectives. In summary, while the values obtained cannot be directly applied to other contexts without first undergoing an evaluation, the methodology described in the manuscript to determine the parameters is replicable.

In this context, the intention of this work was not to validate these specific masks to several datasets but rather to provide a straightforward and user-friendly method for creating and applying masks to PLDs. The main advantages of the presented methodology are its cost-effectiveness and the lack of need for programming experience, making it accessible without any prerequisites. This step-by-step guide may particularly appeal to students engaged in this research topic who are novices in the programming skills typically required to create visual dot noise masks, such as algorithms [9], offering a valuable resource to explore motion perception using PLDs.

While we have aimed to make this methodology accessible and user-friendly, we recognise that any new approach, regardless of its simplicity, carries the risk of failure or limitations. The effectiveness of this methodology can be evaluated through careful implementation and testing in each specific context. As such, we recommend that researchers validate and adapt the parameters as needed for their own datasets and research goals, ensuring the necessary experimental rigour is maintained.

Finally, we encourage other researchers and animators to explore this and similar platforms further to expand the scope of the provided guidelines. These guidelines are currently limited to a specific type of stimuli, as we focused on human action perception, but they may have broader applicability. Future research could examine these masks in domains beyond human motion. For example, this approach could be adapted for studying human analysis of animal motion, such as gait or flight patterns, as well as for experiments testing whether animals perceive biological motion similarly to humans [28]. Additionally, these masks could be useful for object identification studies to explore how humans or AI systems detect moving objects under visually noisy conditions. The flexibility of the masks, coupled with the accessibility of the software, positions this method as a useful tool for exploring numerous possibilities in perception experiments across diverse contexts.

## Figures and Tables

**Figure 1 neurosci-06-00002-f001:**
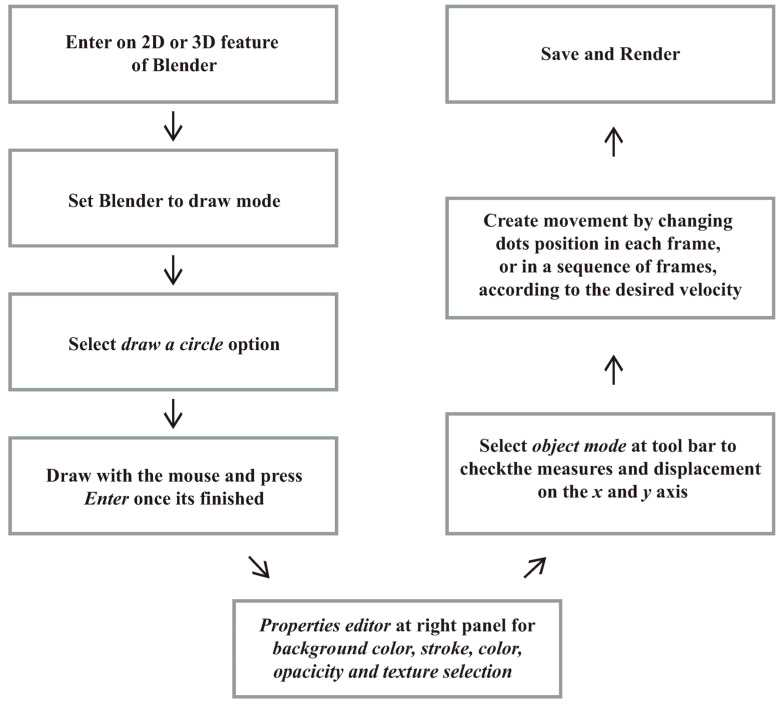
Diagram of the mask-building process. The diagram provides an overview of the step-by-step procedure, highlighting key stages from opening the software to rendering the video.

**Figure 2 neurosci-06-00002-f002:**
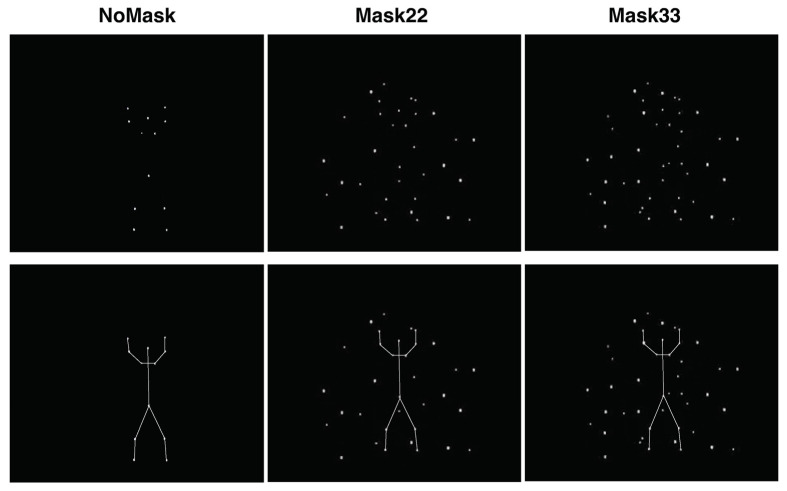
Examples of biological PLDs, specifically frame 15 of the jumping jack action, in the three masking conditions: NoMask, Mask22, and Mask33. The top row illustrations show the video as presented in the task. In contrast, the bottom row depicts the same images but with the dots connected to allow visualisation of the human agent.

**Figure 3 neurosci-06-00002-f003:**
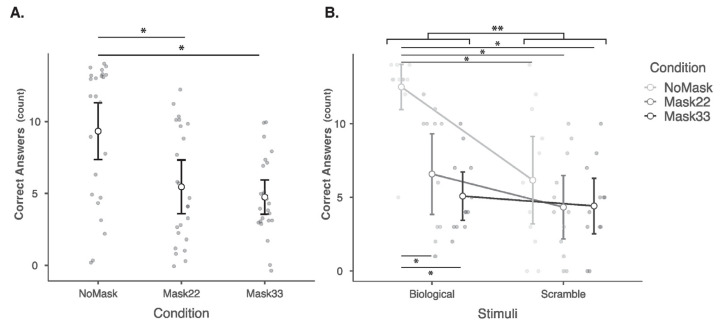
Panel (**A**) shows means and confidence intervals of accuracy according to condition. As can be seen, accuracy for NoMask (9.3 ± 0.897 SE) > Mask22 (5.46 ± 0.849 SE) = Mask33 (4.75 ± 0.542 SE). Panel (**B**) shows means and confidence intervals of accuracy according to stimuli and condition. As can be seen in this panel, accuracy for biological (8.06 ± 0.735) > scrambled (4.97 ± 1.001) stimuli. Further, the accuracy for BioNoMask stimuli was significantly higher than for all other conditions. * indicates *p* < 0.001, and ** indicates *p* < 0.05.

**Table 1 neurosci-06-00002-t001:** Mean number of videos with correct response; standard error and confidence interval for each stimuli type.

Condition	Stimuli	Mean	SE	95% CI (Lower/Upper)
NoMask	Biological	12.50	0.702	10.96/14.04
NoMask	Scramble	6.17	1.347	3.20/9.13
Mask22	Biological	6.58	1.246	3.84/9.33
Mask22	Scramble	4.33	0.980	2.18/6.49
Mask33	Biological	5.08	0.743	3.45/6.72
Mask33	Scramble	4.42	0.857	2.53/6.30

**Table 2 neurosci-06-00002-t002:** Percentage accuracy in detecting a human action in each biological stimulus according to condition.

Biological Motion	NoMask (%)	Mask22 (%)	Mask33 (%)
Jumping Jack	95.83	45.83	45.83
Walk	70.83	66.66	45.83
Lateral Step	87.50	58.33	50
March	91.67	50	50
Low Kick	95.83	37.50	20.83
Jump	91.67	45.83	29.17
Jump Rope	91.67	25	91.67

## Data Availability

The files for the masks described herein, as well as the Blender project and the datasets generated and analysed during the current study, available for download as .zip archives from an online repository at https://osf.io/q2u95/, accessed on 31 December 2024.

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
