# Peer review of "Visual Noise Mask for Human Point-Light Displays: A Coding-Free Approach"

_neurosci, 2025, doi:10.3390/neurosci6010002_

Round 1
Reviewer 1 Report
Comments and Suggestions for Authors
This study is worthy of publication; however, several significant issues need to be addressed:
1) In Section 2, please expand on the neural processing pathway: include visual neural response from eye input, neural processing of point-light information, and subsequent motion control.
2) In Figure 1, add formulations within the content to describe the process in detail.
3) In Figure 2, clarify how the noise is generated, specify the noise intensity, and explain how the real signal's point-light is enhanced.
4) The results section could be strengthened with a detailed case study to illustrate the intermediate processing stages.
Reviewer 2 Report
Comments and Suggestions for Authors
In the manuscript, the authors presented the first user-friendly step-by-step guide on developing visual noise to mask PLDs using free open-source software that offers compatibility with various file formats, features a graphical interface, and facilitates the manipulation of both 2D and 3D videos. 12 subjects were used to test the two generated masks, showing the efficiency. It will be interesting for the audiences in the relevant fields. However, there are some issues that authors need to solve before publication.
1. Could authors include the criteria used to determine that the noise masks effectively "jeopardised human agent recognition"? What specific metrics or thresholds did the authors use?
2. How did the authors select the 12 subjects for the pilot experiment?
3. Were there any notable variations in recognition difficulty across different types of human movements or action categories?
4. Authors claimed ease of use for novice researchers. How do authors theoretically situate methodological accessibility against the potential risks of reduced experimental rigor?
Reviewer 3 Report
Comments and Suggestions for Authors
The reviewer understands that Senra et al. presented a manuscript entitled "Visual noise mask for human point-light displays: a coding-free approach". The reviewer has a few questions and they would like to request authors to kindly answer all the questions by updating their manuscript.
1) How were the dynamic masks' specific properties (such as dot sizes and movement intervals) determined? Are these parameters applicable to other types of PLD stimuli?
2) The study employed 22- and 33-dot masks. How generalizable are these figures for masking various PLD configurations? Would more complicated masks (e.g., with varying motion trajectories) offer additional advantages?
3) What was the reason for introducing a block with unmasked stimuli (Block in the pilot experiment? Could this have caused biases in the participants' ability to discern masked stimuli?
4) The pilot trial had 12 people. Was the sample size large enough to identify small-to-moderate effect sizes? How was statistical power calculated?
5) Have the produced masks been validated on other datasets or with multiple participant groups to ensure consistent performance?
6) What restrictions, if any, did the authors find in utilizing Blender for masking? For example, were there limitations in file formats or rendering times that may prevent its widespread adoption?
7) How relevant is this masking approach to other areas than human action perception, such as animal motion analysis and object identification studies?
8) Please give units for the x and y axes. Provide all figures in HD format.
9) The manuscript claims that the masks and associated data would be made available online. What steps are in place to guarantee that these materials are user-friendly and well-documented for research use?
Round 2
Reviewer 1 Report
Comments and Suggestions for Authors
These revisions have been addressed, I believe the paper is suitable for publication.
Reviewer 2 Report
Comments and Suggestions for Authors
The paper can be accepted as is now.
Reviewer 3 Report
Comments and Suggestions for Authors
I accept the updated version of the manuscript. Please add the figures from the past manuscript in the updated manuscript as well.